# Foreign Direct Investment Dynamic Performance with Low-Carbon Influence: A Provincial Comparative Application in China

**Xinna Zhao [1,2]** **, Yuhang Tang [1,*], Milin Lu [1] and Xiaohong Zhang [1,2]**

[1]   School of Economics & Management, Beijing Institute of Petrochemical Technology, 19 Qingyuan Road, Beijing 102617, China
[2]   Enterprise Development Research Center, Beijing Institute of Petrochemical Technology, 19 Qingyuan Road, Beijing 102617, China
*   Correspondence: hibaboy@sohu.com; Tel.: +86-1839-776-7617

**Abstract:** Cross-border capital flows have been a major force driving economic globalization. Foreign direct investment (FDI) plays a decisive role in seeking out market technology brands and enhancing the global competitiveness among international inflows. With the requirement of economic development, this paper focused on a performance evaluation of FDI in China. However, because of the planned transformation to a market economy in China, FDI has been promoted with a regional cascade structure. Similar to the development track of the Chinese economy, it is necessary to evaluate FDI quality more than purely quantity from a provincial point of view. Therefore, this paper evaluated the Chinese provincial FDI total factor productivity using the dynamic Malmquist model. In contrast to traditional evaluations, this paper focused on inter-temporal influence in FDI performance evaluation. To understand the inter-temporal effects, physical capital stock was defined as a dynamic variable in FDI sustainability performance. Additionally, with the pressure to reduce emissions, energy consumption was also considered during the evaluation. The empirical results revealed that the dynamic variable is the bottleneck in FDI performance for most Chinese provinces. It is only efficient in a few municipalities and provinces, such as Shanghai and Guangdong. Additionally, energy conservation was more efficient in the performance evaluation of eastern regions in China.

**Keywords:** FDI; energy conservation; international finance; regional cascade structure; dynamic effect

## 1. Introduction

Over the past few decades, cross-border capital flows have been a major force driving economic globalization. In the long run, financial development and trade openness are more important for developing economies, while financial development with financial openness is more important for advanced economies (Chortareas et al. 2015). Foreign direct investment (FDI) plays the decisive role in resource allocation within the market (Dal Bianco and Loan 2017). It seeks market technology brands and enhances the global competitiveness among international inflows. Thus, FDI is essential for economic development in China. While global FDI has declined by 13%, Chinese FDI inflows showed an increase of 4% in 2018. In this, China ranked second in the world, after only the United States. A total of 60,533 foreign-invested enterprises have been established nationwide in 2019, an increase of 69.8% compared to 2018. FDI from the UK increased by 150.1%, FDI from Germany increased by 79.3%, and that of the United States increased by 7.7%.

However, because of the planned transformation to a market economy, the Chinese economy has been promoted with a regional cascade structure. Subsequently, FDI in China has been bolstered with the same structure. Thus, FDI in China is disproportionately distributed. Similar to some other counties, such as Poland, regional differences originate from resource endowment, path dependency based on the locations of industry and big city agglomerations, infrastructure quality, different economic prospects of potential cooperation with neighboring countries, or transformation challenges (Nazarczuk and Umiński 2018). Additionally, similarly to the development track of the Chinese economy, FDI quality is more essential than quantity alone, especially in special economic zones (SEZs). To understand the inter-temporal effects, FDI sustainability performance is evaluated depending on dynamic factors, such as the physical capital stock. Additionally, evaluation of the dynamic performance of provincial FDI in China should also consider energy consumption and the pressure to reduce carbon emissions. Therefore, this paper focused not only on the traditional performance of provincial FDI in China, but also considered energy influence from the perspective of a low-carbon economy. Moreover, physical capital stock as the inter-temporal effect was used to make the dynamic performance more generic.

A literature review revealed the importance of FDI performance (Rashid and Lin 2018). Saidi et al. (2018) highlighted that FDI had a positive effect on economic dimensions, including GDP (Gross Domestic Product) growth, technical progress, and management improvement. Researchers have explored the important role of FDI in economic efficiency improvement (Belloumi and Alshehry 2018). Therefore, FDI is very important for economic development. Nourzad (2008) indicated that the driving force from FDI is expressed via multiple channels, such as technical transfer, providing impetus for the economy. Previous studies have compared the differences in FDI performance among provinces. Kueh (1992) discussed the reasons for FDI distribution diversity and the implications of policy support. The results suggested that FDI distribution changes from the southeast to the north and northeast coasts, as well as in the midland and western region.

In the context of a stabilized economy and increasing FDI, previous studies have focused on the evaluation variables for FDI, such as the degree of openness (Walter and Ugelow 1969), domestic investment (Cheng and Kwan 2000), the labor force (Lei et al. 2013), and the allocation of resources (Rashid and Lin 2018). Additionally, extensive economic development increases resource usage and environmental pressure. From a sustainability perspective, FDI, as a critical driving force of economic development, also can be influenced by energy consumption. Previous studies have indicated that weak environmental regulations and abundant energy sources are an advantage to FDI in developing countries. In addition, Hübler and Keller (2010) identified a positive correlation between FDI and improved local energy efficiency.

The literature review revealed that the Malmquist model has been widely used to evaluate total factor productivity (TFP), such as regional technological innovation efficiency (Wang et al. 2012; Hou 2016), regional financial efficiency (Yao et al. 2016; Liu et al. 2018), green economy performance (Yang et al. 2015; Pérez et al. 2017), and FDI performance (Liu et al. 2016). The input and output variables in these studies are always selected to reflect the cost sources as the input variables and revenue sources as the output variables (Feng and Serletis 2010; Portela and Thanassoulis 2010; Jiang and He 2018). However, some special variables not only take effect in one period, but also have the long run effect, such as deposits in banks and fixed assets in companies (Bagchi et al. 2019). As Zhao and Zhong (2017) pointed out, a traditional Malmquist model does not account for the effects of these special variables between consecutive periods. This can lead the evaluation results to over-incentivize in the short run and under-incentivize in the long run. The dynamic DEA (Data Envelopment Analysis) model was proposed for dealing with inter-connected activities by Färe and Grosskopf (1996). Tone and Tsutsui (2010) extended the dynamic DEA model by adding carry-over effects among different periods in the framework of the SBM (Slacks-based Measure) model, called Dynamic SBM (DSBM). For the inter-temporal effect of multiple periods, this paper

introduced the dynamic thinking from DSBM into a traditional Malmquist model to build a dynamic Malmquist model.

This paper has made contributions at the methodological and application levels. First, this paper has built the dynamic Malmquist model considering the physical capital stock as the inter-temporal effect into the traditional Malmquist model. This paper has facilitated explaining the sustainability of participating in provincial FDI considering the physical capital stock as carry-over activities. Carry-over activities are defined as the special variables which not only take effect in one period, but also have a long run effect in subsequent periods. Second, considering the low-carbon economy, this paper has evaluated FDI performance with the energy consumption influence. Third, this paper has investigated the different contributions of evaluation variables to show the regional differences of FDI. Additionally, this paper also has practical implications. The empirical results provide decision makers with a deeper understanding about the FDI sustainable development mechanism and investors with a reference of FDI distribution.

This paper is structured as follows: First, a conceptual dynamic Malmquist model is constructed and decomposed. The measurements of provincial FDI performances are presented in Section 2. Then, from a dynamic point of view, an example of provincial FDI in China is discussed using cluster analysis to analyze evaluation variables contribution in Section 3. Lastly, the conclusions as well as the suggestions and limitations are presented in the last section.

## 2. Methodology Model and Data Explanation

### 2.1. Decomposing the Dynamic Malmquist Model

Before describing the detailed equations, the conceptual framework of the dynamic Malmquist model was built, as shown in Figure 1. Provinces in China are defined as the decision-making units (DMUs) which are the objects of this study. Färe et al. (1994) proposed a best-practice frontier and decomposed Malmquist model to evaluate the different efficiency indices of product feasible sets. At period $t$ ($t = 1, 2, 3 \dots$ ), DMUs produce output $Y^t$ (desirable output $Y^{dt}$ and undesirable output $Y^{ut}$) using input $X^t$ and dynamic variable $Z^t$. The carry-over activity $Z^t$ ($t = 1, 2, 3 \dots$ ) connects the adjacent periods $t$-1 and $t$.

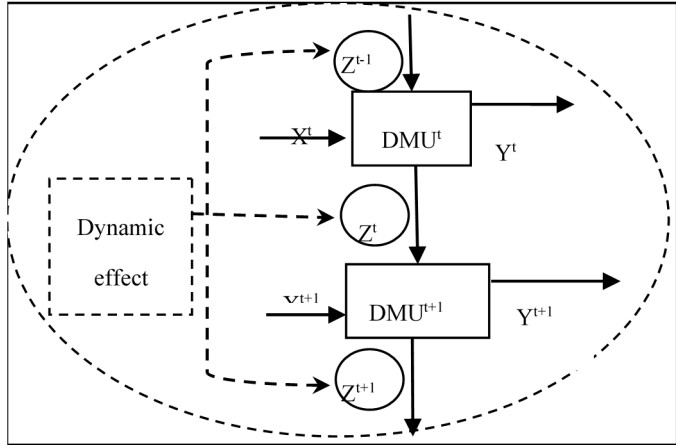

**Figure 1.** Dynamic Malmquist model structure.

Thus, the product feasible sets of DMUs at period $t$ are shown as Equation (1):

$$
\begin{aligned}
&\forall \left( x^t, z^{t-1}, y^t, z^t \right) \in S_t \\
&\equiv D^t\left( \left( x^t, z^{t-1}, y^t, z^t \right) \Big| \left( X^t, Z^{t-1}, Y^t, Z^t \right) \right) \triangleq \sup\left\{ \theta : \left( x^t, z^{t-1}, \theta y^t, \theta z^t \right) \in S^t \right\}
\end{aligned}
\tag{1}
$$

In Equation (1), sup means the lowest upper bound. The distance function $D^t$ is the DMUs' efficiency at period $t$ with respect to the efficient frontier that considers the inter-temporal effect. Thus, $D^t\left(\left(x, z^{t-1}, y, z^t\right) \big| \left(X^t, Z^{t-1}, Y^t, Z^t\right)\right)$ equals the efficiency of DMU(x, $z^{t-1}$, y, $z^t$), comparing with that of DMU($X$, $Z^{t-1}$, $Y$, $Z^t$), which is on the best practice frontier.

Similarly, the DMUs' efficiency at period $t + 1$ are shown as Equation (2):

$$\forall \left(x^{t+1}, z^t, y^{t+1}, z^{t+1}\right) \in S_{t+1}$$
$$\equiv D^{t+1}\left(\left(x^{t+1}, z^t, y^{t+1}, z^{t+1}\right) \big| \left(X^{t+1}, Z^t, Y^{t+1}, Z^{t+1}\right)\right) \triangleq \sup\left\{\theta : \left(x^{t+1}, z^t, \theta y^{t+1}, \theta z^{t+1}\right) \in S^{t+1}\right\} \tag{2}$$

Practically, the distance function $D^{t+1}$ is the DMUs' efficiency at period $t + 1$, with respect to the efficient frontier considering the inter-temporal effect.

To identify the influence from other DMUs, evaluation variables of one DMU at period $t + 1$ are defined as effective variables at period $t$. Based on this thinking, evaluation variables at period $t + 1$ are defined by comparing them with the other DMUs at period $t$. This distance function is shown as Equation (3):

$$if \left(x^{t+1}, z^t, y^{t+1}, z^{t+1}\right) \in S^t$$
$$\Rightarrow D^t\left(\left(x^{t+1}, z^t, y^{t+1}, z^{t+1}\right) \big| \left(X^t, Z^{t-1}, Y^t, Z^t\right)\right) \triangleq \sup\left\{\theta : \left(x^{t+1}, z^t, \theta y^{t+1}, \theta z^{t+1}\right) \in S^t\right\} \tag{3}$$

On the thinking of traditional Malmquist decomposition, the dynamic Malmquist model is also decomposed into two parts. The traditional model is extended into the new model considering carry-over activities. Thus, dynamic Malmquist can be composed into overall efficiency change (DTEC) and dynamic technical change (DTC). Similar to traditional Malmquist decomposition thinking (Färe et al. 1994), DTEC means the attractive power for performance improvement, such as the allocation of evaluation variables and scale efficiency. Additionally, DTC is denoted as the potential power, such as the managerial style.

## 2.2. Data Explanation

Previous studies on performance evaluation have always focused on physical capital stock, human capital stock, and output value based on the economic growth theory. Additionally, the hysteresis of physical capital stock existed in economic growth and technical progress (He 2001). Furthermore, Hou (2016) proved the objectivity of physical capital stock hysteresis. He has also demonstrated that GDP growth depended on the previous year's fixed capital investment. This means the sustainability of total factor productivity is not only represented by the congruence of evaluation variables, but also reflects in the sustainability of physical capital stock.

Similar to most of the previous studies, average FDI is defined as an output variable. Average human capital stock, average energy consumption, and export rate are defined as input variables. Considering the hysteresis of physical capital stock, average physical capital stock is defined as a dynamic variable. For simple calculation, the inter-temporal effect of physical capital stock only focuses on the adjacent periods. The definitions of evaluation variables are shown as follows:

1. Physical capital stock: On the thinking of the perpetual inventory method, first, the current fixed capital formation is defined as the current investment. Then, the basic period is defined as 10 times the fixed capital formation at 1952. Last, the depreciation rate as 10.96% is used during the calculation, based on Lei (2009).

2. Human capital stock: Educated manpower is the main inducement among FDI competitions. Based on the thinking of Zhao and Zhang (2009), this paper denoted illiteracy as 3, primary education as 6, junior high school education as 9, high school education as 12, and college education and above as 16. The sum of the indices is defined as human capital stock.

3. Energy consumption: The energy cost and availability are the main factors considering the low-carbon economic development. For some missing data, the average data of adjacent

provinces is offered. For example, the data of the Hainan province at 2002, and the data of the Hunan province at 1997 and 1998.

4.  Export rate: The economic openness is a key factor for the multinationals investment. The regional openness is a basic factor of FDI performance. So, the export rate is defined as the openness benchmark based on Nazarczuk and Umiński (2018).

5.  Foreign directed investment: FDI is defined as a production function. Using the same input variables, the more funds indicates the more investment benefits.

For eliminating the influence of provincial population differences, average data is used in the empirical analysis. For avoiding the multicollinearity problem, the correlation coefficients of all evaluation variables are listed in Table 1. Similar to statistics rules, the correlation coefficient of 0.75 is a high one. So, an asterisk is added for the significant correlations in Table 1.

**Table 1.** Correlation coefficients of evaluation variables.

| Correlation Coefficient | Human Capital Stock | Energy Consumption | Export Rate | Physical Capital Stock | FDI |
|---|---|---|---|---|---|
| Human Capital Stock | 1 | 0.557 | 0.492 | 0.750 * | 0.666 |
| Energy Consumption | 0.557 | 1 | 0.322 | 0.638 | 0.525 |
| Export Rate | 0.492 | 0.322 | 1 | 0.519 | 0.734 |
| Physical Capital Stock | 0.750 * | 0.638 | 0.519 | 1 | 0.677 |
| FDI | 0.666 | 0.525 | 0.734 | 0.677 | 1 |

* stands for the significant correlations.

The results reveal that the range of indices is from 0.3 to 0.8. This indicates a correlation among the evaluation variables. Additionally, the output variable is significantly correlated with the other variables. This proves the validity of the evaluation framework for examining the data.

All of the data are from the official statistics, such as China statistical yearbook, China city statistical yearbook, and China energy statistical yearbook. As shown in Table 2, the descriptive statistics results of evaluation variables are shown as the descriptive statistics, such as average, standard deviation (S.D), minimum, and maximum.

**Table 2.** Descriptive statistics of the variables used in the study.

| Factors | Sample | Average | S.D | Min | Max |
|---|---|---|---|---|---|
| Human Capital Stock | 510 | 8.525 | 1.080 | 5.930 | 13.329 |
| Energy Consumption | 510 | 2.349 | 1.474 | 0.065 | 7.947 |
| Export Rate | 510 | 0.158 | 0.187 | 0.015 | 0.864 |
| Physical Capital Stock | 510 | 1.441 | 1.852 | 0.030 | 12.247 |
| FDI | 510 | 95.660 | 145.882 | 0.498 | 1143.109 |

As shown in Table 2, input variables exhibit minimal differences. However, FDIs are quite different. This indicates that dynamic FDI performance exhibits significant differences among provinces using the relatively fixed inputs. Therefore, it is necessary to seek for the bottlenecks in dynamic FDI performance improvement. So, the rest of the paper is a thorough analysis on the differences of provincial FDI performances.

## 3. Empirical Results and Discussion

### 3.1. Heterogeneity of Foreign Directed Investment (FDI) Dynamic Performance in China

Dynamic FDI performance in China is denoted as dynamic total factor productivity (dynamic Malmquist productivity index, DMPI) in Figure 2. The diachronic results from 1997 to 2013 indicate that dynamic FDI performance exhibits a rising trend with fluctuations in China. The fluctuations

subsequently became less pronounced after 2004. Compared with the benchmark of 1, DMPI was 1.08 in China during the whole period. This indicates a small progress of dynamic FDI performance in China. To obtain a diachronic perspective, the whole period is divided into two stages, with 2004 as a boundary. In the first stage, from 1997 to 2004, DMPI exhibits a "W"-shaped curve with up-and-down cycles for four years. The two slight rises indicate that DMPI in 2004 peaked for the whole period. Subsequently, the DMPI curve became relatively flat with a slight decline. Additionally, DMPI in 2012 was less than 1. Although DMPI increased in 2013, the trend remained declining overall. The trend of DMPI indicates dynamic FDI performance. This means the sustainability of evaluation variables does not correspond with the economic development. Therefore, it is necessary to analyze the bottlenecks of dynamic performance development in China.

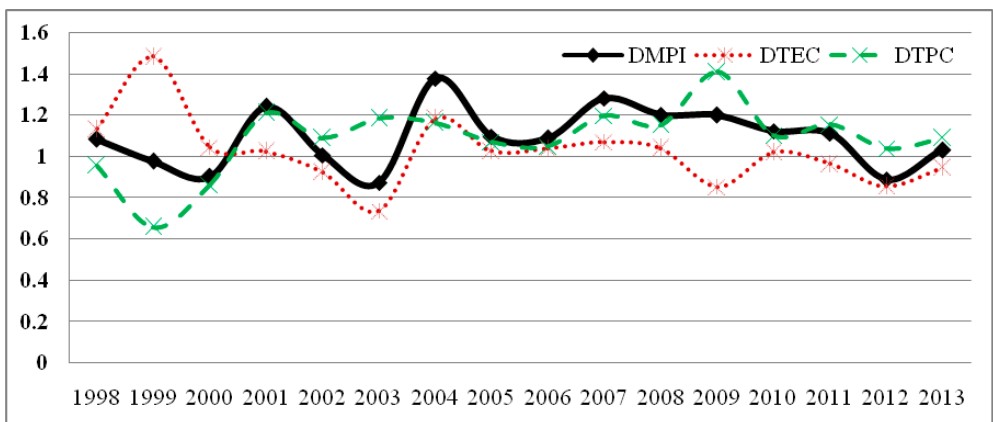

**Figure 2.** Decomposing foreign directed investment (FDI) dynamic performance in China.

As shown in Figure 2, both DTEC and DTC are determinants of DMPI. Therefore, the attractive power is denoted by DTEC while the potential power is denoted by DTC. The two indices are critical to the total factor productivity of FDI in China. However, the influences of the indices are significantly different. In the first two years, DTEC substantially contributed to DMPI, whereas DTC became the major factor for improvement in DMPIs after 2000. This suggests that current changes in potential power, such as employee quality and advanced management, have been a big step forward for provincial FDI performance. It is dangerous to make effort to attract a single performance. This means all the resources are put into a single operation without considering the subsequent influence. This retrogression means the inter-temporal effect is gradually ignored. This ignored phenomenon leads provincial FDI to give up opportunities in the long run for the special benefit in the short run performance.

Thus, the dominant determinant of FDI performance is quality efficiency improvement with the promotion of technical development. The allocation efficiency and scale efficiency contributed to the stable basic in the first two years. However, DTEC was lower than DMPI after 2000. This implies that the allocation efficiency and scale efficiency change into a constraint to performance improvement. For example, DTEC in 2000 tended to decrease. Additionally, that in 2009 was less than 1, which means to retrogress. With the policy bonus of China, such as Western Development in 2000 and the Four Trillion Yuan Stimulus Plan in 2009, decision makers reinforced the government investment to attract FDI. However, they have focused on the quantity and ignored the quality without control of variable efficiency.

DTEC also constrains dynamic FDI performance, as shown in Figure 2. DTEC can be further composed into DPTC (dynamic pure technology efficiency change) and DSEC (dynamic scale efficiency change) as shown in Figure 3. This implies that the allocation efficiency of FDI variables is crucial for the attractive power. Based on the decomposition of the Malmquist model, the allocation efficiency (DTEC) can be further decomposed into technology efficiency (DPTC) and scale efficiency (DSEC). DPTC in 2000 initially tended to decline, combined with a continued declining tendency of DSEC since

2000. This result is consistent with those of previous studies on allocation efficiency in 2000, with the influence of Western Development policy in China.

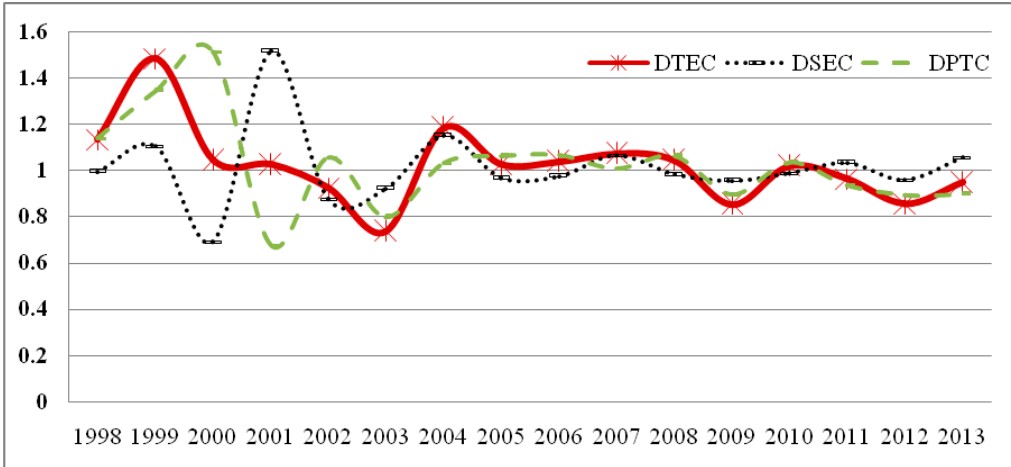

**Figure 3.** Decomposing dynamic technology efficiency changes in China.

The provincial diachronic result from 1997 to 2013 highlights the differences in dynamic FDI performance in China. As shown in Figure 4, the deeper color province indicates the higher FDI performance. The different color provincial boundary shows the different regions in China. The white boundary is midland in China. The brighter boundary is the western province in China and the deeper one is the eastern province in China. Using a regional perspective, the advantaged provinces always belong to the western region. All the efficiency indices are greater than 1. This indicates that provincial FDI performance always progresses during the evaluation periods. The improvement of dynamic FDI performance in the western provinces was facilitated by the policy bonus of Western Development in China.

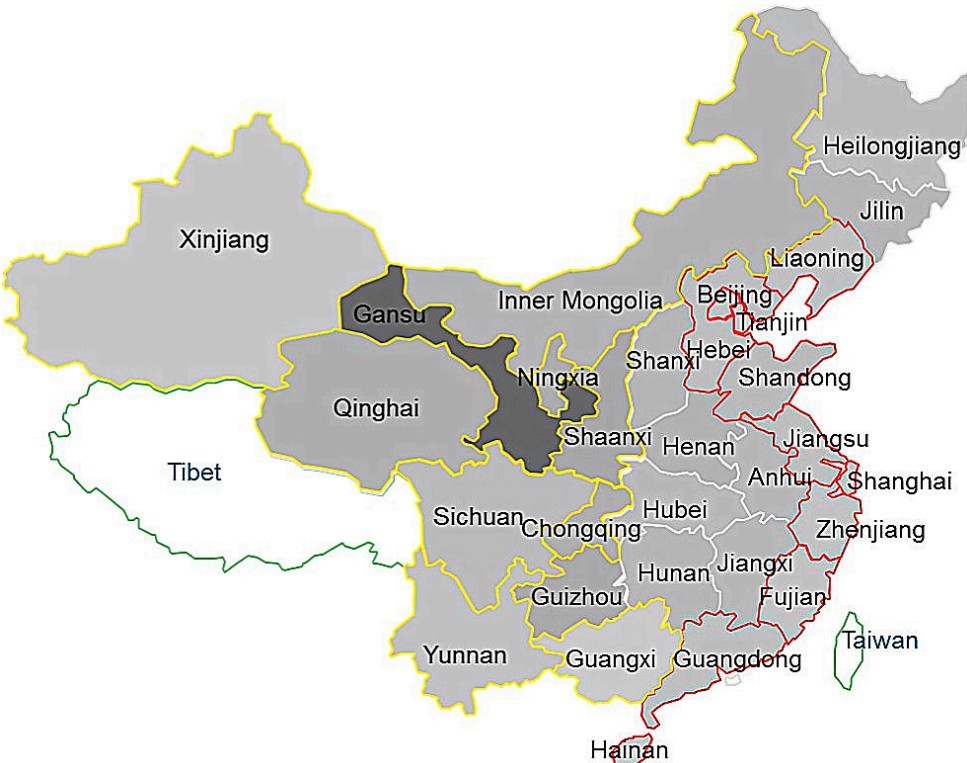

**Figure 4.** Dynamic performance distribution in China.

The first three advantaged provinces and the last three disadvantaged provinces are shown in Table 3 according to the performance indices. For the decomposed efficiency indices, DTC significantly enhances dynamic FDI performance. This result proves that an indirect constraint on FDI performance is the attractive power. Therefore, the effective use of evaluation variables is the solution to promoting performance. For example, Xinjiang and Guangxi in the western region are disadvantaged provinces of dynamic FDI performance. The decomposition is used in order to identify the critical restriction variables. First, from the allocation efficiency perspective, all the advantaged provinces are good at DPTCs in the western region, such as Inner Mongolia, Ningxia, and Guizhou. However, the disadvantaged provinces are in different regions, such as Fujian in the eastern region, Heilongjiang in the midland region, and Xinjiang in the western region. Thus, regional differences are not a critical constraint. This proves the proportionality of macro-control. Additionally, from a perspective of scale efficiency, the advantaged provinces of DSECs are Qinghai, Jiangxi, and Guizhou. While the disadvantaged provinces are Guangxi, Hainan, and Gansu.

**Table 3.** Provincial FDI performance rank.

| Rank | Advantage Province | | | Disadvantage Provinces | | |
|---|---|---|---|---|---|---|
| **Province** | **Gansu** | **Shaanxi** | **Guizhou** | **Hainan** | **Xinjiang** | **Guangxi** |
| DMPI | 1.803 | 1.235 | 1.216 | 0.976 | 0.955 | 0.932 |
| Province | Gansu | Shaanxi | Xinjiang | Liaoning | Yunnan | Inner Mongolia |
| DTC | 1.976 | 1.197 | 1.179 | 0.97 | 0.967 | 0.96 |
| Province | Inner Mongolia | Guizhou | Qinghai | Hainan | Gansu | Xinjiang |
| DTEC | 1.167 | 1.098 | 1.078 | 0.929 | 0.913 | 0.81 |
| Province | Inner Mongolia | Ningxia | Guizhou | Heilongjiang | Fujian | Xinjiang |
| DPTC | 1.113 | 1.059 | 1.038 | 0.968 | 0.968 | 0.827 |
| Province | Qinghai | Jiangxi | Guizhou | Guangxi | Hainan | Gansu |
| DSEC | 1.078 | 1.077 | 1.058 | 0.949 | 0.929 | 0.913 |

DMPI stands for dynamic Malmqusit productivity index. DTC stands for dynamic technical change. DTEC stands for overall efficiency change. DPTC stands for dynamic pure technology efficiency change. DSEC stands for dynamic scale efficiency change.

In conclusion, the solution of dynamic FDI performance improvement is the combined promotion of multiple decomposed efficiency indices. Additionally, policy bonus is beneficial to FDI performance. However, the key is realizing the sustainability of the effects from the policy bonus.

### 3.2. Hierarchical Clustering of Provincial FDI Performance

To identify the interprovincial cooperative effect for dynamic FDI performance, all the provinces are classified using hierarchical clustering. For the previous calculation by the dynamic Malmquist model, the different performances of provincial FDI include the influence of the regional resource endowments. Therefore, three decomposed efficiency indices (DTC, DSEC, and DPTC) with the regional heterogeneity are the basics for the cluster analysis. Additionally, several other methods are tested including the Ward-method based on the theory of hierarchical cluster analysis. Only the furthest neighbor distance method can classify the discriminations of provincial FDI performance. For example, based on the Ward-method, cluster one includes 16 provinces which are more than one half of Chinese provinces. Both of cluster two and cluster four include three provinces. So, the furthest neighbor distance method is more suitable for this empirical analysis compared with other methods.

Based on the provincial regional resource endowment, all the provinces in China are classified into four clusters, as shown in Figure 5. From a perspective of regional divide, the western provinces are classified into cluster two and cluster three. The midland provinces are classified into cluster three and cluster four, while the eastern provinces are classified into cluster one and cluster two. The regional influence is significant for the decomposed efficiency indices of FDI performance.

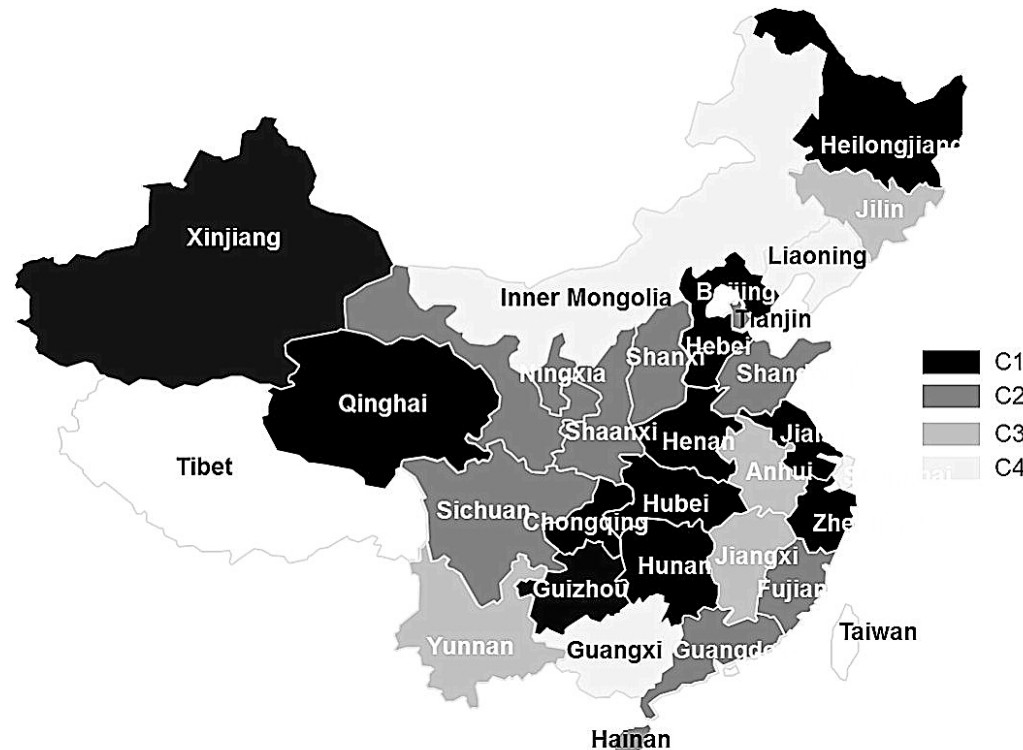

**Figure 5.** Cluster analysis distribution of FDI dynamic performance.

Based on cluster analysis, the average decomposed indices and the corresponding rankings are shown in Table 4. The potential power effect is denoted by DTC. The allocation effect is denoted by DPTC. While, the scale effect is denoted by DSEC. Based on the thinking of the BCG Matrix (Boston consulting group matrix), the four clusters can be named as Dog group, Question Mark group, Cash Cow group, and Star group.

**Table 4.** Average decomposed indices of cluster analysis.

| Province | DTC | DPTC | DSEC |
|---|---|---|---|
| Cluster 1 | 0.977 | 1.037 | 1.003 |
| Cluster 2 | 1.177 | 1.006 | 0.975 |
| Cluster 3 | 1.077 | 0.983 | 1.025 |
| Cluster 4 | 1.008 | 1.009 | 1.054 |

Cluster one has an absolute advantage for the first rank in DPTC. The allocation efficiency of FDI variables is the basis of the advantage. The dynamic FDI performance of cluster one is lower than that of the other clusters. Although cluster one is good at the basic efficiency index as a location of all the variables. The average of DTCs is less than 1. Therefore, it is a bottleneck (DTC = 0.977) for the sustainability development. The lowest rank of DTC indicates the absence of potential power control, such as sustainability of policy bonus and incorporation of advanced management. From a perspective of provinces in China, the municipalities, such as Beijing and Shanghai, belong to cluster one. Their decomposed efficiency indices exhibit a relatively flat trend around the benchmark of 1 during the whole period. The municipalities attract FDI at earlier years. They also had a higher degree of openness. Therefore, the attractive conditions of FDI are relatively mature. Nevertheless, the policy bonus is weaker than before. So, the performance of municipalities gradually regresses. Because of this bottleneck, FDI improvement in these regions is much more difficult than that of other regions. Inner Mongolia is a special province in cluster one for exhibiting an alternate trend. At the beginning of the 1990s, Inner Mongolia faced difficulties to attract FDI. The DTEC improved rapidly because of the

lower basic status. Therefore, an absolute advantage in the allocation efficiency of evaluation variables makes Inner Mongolia a leader in the western region. Additionally, the regional advantage is beneficial to FDI because Inner Mongolia borders Central Asia with more policy bonus of One Belt One Road.

With a leading position in allocation efficiency, the main evaluation criterion presents profit growth. So, the provincial FDIs in cluster one are similar to the Dog group based on the thinking of the BCG Matrix. Based on the efficiency indices, the strategy of cluster one is stabilization development. The most efficient breakthrough for improving performance is keeping the advantage of pure economic growth in traditional evaluation. For more economic development, this strategy is to give full advantage of inter-temporal activity. This influence cannot be ignored.

Cluster two has an absolute advantage in DTC for the first rank in dynamic performance. These provinces take advantage of the sustainability for an advantage of potential power. Nevertheless, the average of DSECs (0.975) is a major concern. The retrogressive scale effect hinders the performance development. For example, the DSEC of Gansu (DSEC = 0.913) represents a typical province in cluster two. With the policy bonus of China Western Development, the DTPC of Gansu (DTPC = 1.9758) is much higher than other provinces in cluster two. Nevertheless, Gansu cannot use the extensive assistance provided which results in the scale effect problem.

With the obvious advantage of potential power, the strategy of cluster two is expansion. The most efficient breakthrough strategy for improving scale efficiency growth in a long run performance. Based on the thinking of the BCG Matrix, cluster two can improve the performance to strengthen a leading position as a typical Star group. Considering the sustainability, on one hand, cluster two should evaluate the inter-temporal efficiency changes of carry-over activity, such as physical capital stock in a long run development. On the other hand, cluster two should continue to consider attractive power strengthen to remedy the pure economic data without considering the completeness of outputs. In conclusion, the traditional advantages with a sustainability point of view can keep cluster two in a top position.

Cluster three has a disadvantage in DPTC which is the basic efficiency index for performance. This represents the allocation efficiency of FDI variables. The retrogressive DPTC influences other efficiency indices. Another threat is the lack of an absolute advantage according to the results of decomposed efficiency indices. For example, similar to Xinjiang, Heilongjiang has a regional advantage because it borders Central Asia. Heilongjiang has capitalized on the opportunities of One Belt and One Road. Additionally, the allocation of evaluation variables is remarkably inefficient. Cluster three is the largest cluster with generality concerns. Most of the provinces blindly pursue the quantity increase. This means, emphasis on quantity while neglecting quality constitute a bottleneck for FDI performance.

The strategy of cluster three should be retrenchment. This means they should gather strength to break through in order to avoid changing into a typical Dog group. The most efficient breakthrough for improving performance in this cluster is the control of carry-over activities. This means that strengthened control of carry-over activity, such as physical capital stock, can enhance dynamic FDI performance. The specific method is incorporating the inter-temporal influence into current incentive mechanisms. The embodiment is that the evaluation has not only focused on one period, but also on the inter-temporal period. The special dynamic variables are multiple, such as inter-temporal benefit increasing rate, average daily balance, and average daily contribution. This breakthrough can not only increase the advantage, but also match the strategy of sustainable development.

Cluster four has an absolute advantage in DSEC. Additionally, all the other efficiency indices exhibit an increasing trend. This implies that the indices are higher than 1. Because of the inadequate DTCs, the performance of cluster four is in the last ranking. This means the dynamic FDI performance of cluster four is the lowest. However, Yunnan is an exception to cluster four for an absolute advantage of DTEC. The investors in Yunnan are interested in the objective factors as labor-intensive and low-technology industries. For example, Yunnan has lower price levels, lower energy thresholds, and lower human capital stock. Policy bonus has resulted in considerable improvement on the dynamic

FDI performance of Yunnan. However, from a diachronic perspective, the potential power of dynamic FDI performance in Yunnan is still lower than that of other provinces.

With an advantage position for the scale efficiency, cluster four sets a better example for other clusters. However, all the provinces in cluster four still have problems that need to be solved urgently with the traditional pure economic competition. The strategy of cluster four is reinforcement. The most efficient breakthrough strategy is modernizing the management to develop profit markets rapidly to catch up with other competitors. For example, using think-tank-management-mode transforms traditional pure economic benefit into traditional pure economic efficiency for a higher traditional index, such as TC (technical change) and PTC (pure technology change). If cluster four systematically selects strategies to pursue pure economic growth according to the strength and advantage, cluster four will change into the Star group.

### 3.3. Contribution Analysis of FDI Evvaluation Variables

Dynamic FDI performance in China is affected by the potential power, allocation, and scale of variables. This analysis focuses on the contribution of FDI evaluation variables. Additionally, the variables of Chinese provinces are evaluated using the non-radical and non-oriented slacks-based measure model.

The diachronic results are divided into two stages basing on the different trends in dynamic FDI performance. At the first stage, the average of DMPI exhibits a "W"-shaped curve with up-and-down cycles for four years. At the second stage, the average of DMPI flattens relatively with a slight decline. The average of DMPI declines to lower than 1. This indicates performance regression in 2012. Therefore, three periods of the first stage, the second stage, and 2012 are compared to reveal the different contributions of evaluation variables. The evaluation variables are human capital stock, energy consumption, the export rate, and physical capital stock. The contributions are relatively different during performance evaluation. From a regional perspective, the different contributions of Chinese provinces are shown as in Figures 6–8. The cumulative percentages of contributions are plotted in a cumulative histogram with the ranking from left to right which indicates those of the first stage, the second stage, and 2012. The provinces are ranked according to the average contribution of evaluation variables.

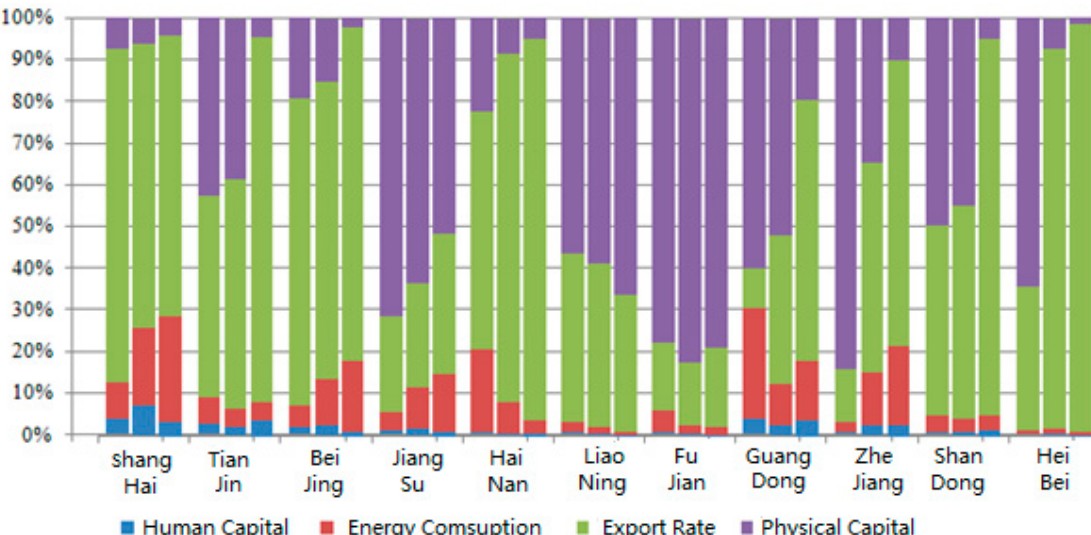

**Figure 6.** Contribution of variables to FDI performance in the east region.

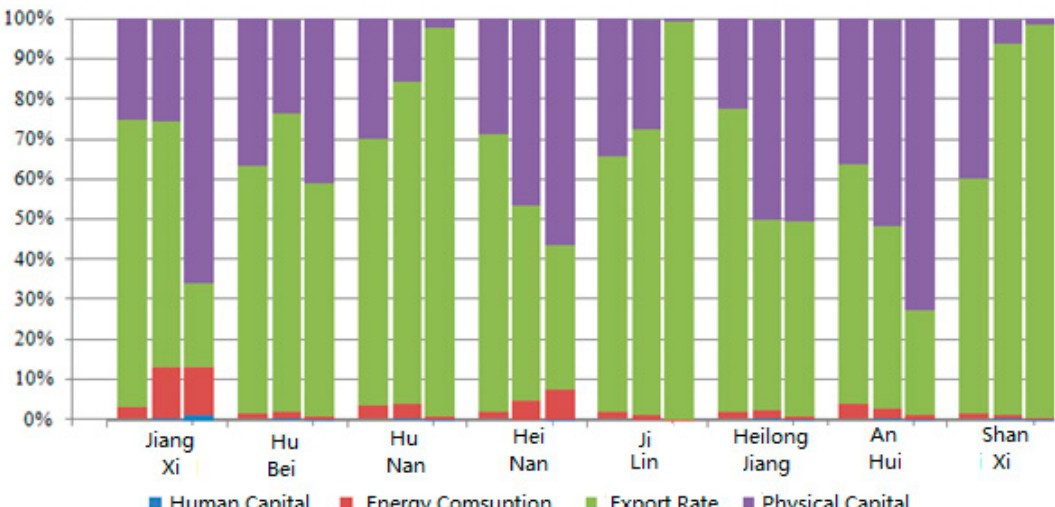

**Figure 7.** Contribution variables to FDI performance in the midland.

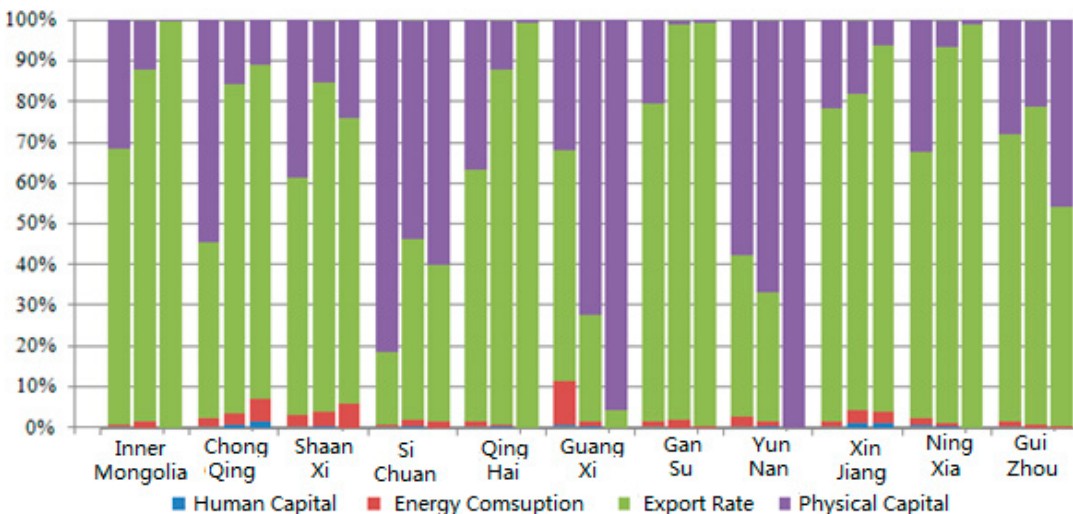

**Figure 8.** Variables contribution of FDI performance in the west region.

The contribution scores of evaluation variables in the eastern region are the highest of three regions in China. As shown in Figure 6, the contribution scores range from 0.43 to 1. The contributions of physical capital stock and export rate provide an absolute advantage in FDI performance, whereas energy consumption rate is relatively lower. Human capital stock has the lowest contribution scores. It is useful in few municipalities and provinces, such as Shanghai and Guangdong.

During the whole period, the contribution of physical capital stock is considerably weak, whereas that of the export rate strengthens. The contribution of human capital stock has an increasing trend. When dynamic FDI performance declined in 2012, the contribution of physical capital stock was gradually replaced by that of the export rate. The three municipalities, including Shanghai, Tianjin, and Beijing, have absolute advantages in FDI performance because of the higher contribution scores. From a contribution perspective, the contribution of the export rate has an absolutely increasing trend. While, that of physical capital stock has a significantly decreasing trend.

The contribution characteristics of Hainan are relatively similar to that of the three municipalities. For the economy of the Beijing-Tianjin-Hebei metropolitan region, the contribution of physical capital stock is weak. Even the region has a decreasing trend. Compared with the decreasing trend, the contribution scores of export rate increased substantially. The Yangtze River Delta, the Pearl River

Delta, and the Pohai Economic Circle have developed with the new industrialization. Therefore, the contribution of physical capital stock is advantageous, though the increase trend is slow.

The contribution scores of evaluation variables in the midland are between the western and eastern in China. As shown in Figure 7, the range of contribution scores was from 0.29 to 0.88. The contribution of the export rate provides an absolute advantage in FDI performance, whereas that of physical capital stock was relatively lower. Energy consumption and human capital stock have the lowest contribution scores. Additionally, the contribution of energy consumption and human capital stock in the midland were also weaker than that of the eastern region.

During the whole period, the contribution had a stable trend. However, the maximum fluctuations were caused by energy consumption, although the contribution of energy consumption increased from 2% to 3%. In contrast to the variance rate of evaluation variables, the contribution of human capital stock changes was much bigger, with a variance rate of 33%. The variance rate of the export rate and physical capital stock were 1% and 3%, respectively. When dynamic FDI performance declined in 2012, the contribution trend in the midland was different from that in the eastern region. This implies that the contribution of export rate was replaced by that of physical capital stock. From a provincial perspective, Jiangxi, Hubei, and Hunan, which border the western region, have an absolute advantage in FDI because of their higher contribution scores (0.9, 0.7, and 0.6, respectively). However, Anhui and Shanxi, which border the eastern region, have lower contribution scores (0.4 and 0.3, respectively). Thus, the midland does not receive a free ride from the eastern region for FDI performance improvement.

The contribution scores of evaluation variables in the western region were the lowest among the three regions in China. As shown in Figure 8, the range of contribution scores was from 0.09 to 0.8. The contribution of export rate provides an absolute advantage in FDI performance. Whereas those of energy consumption and human capital stock were relatively lower.

During the whole period, the contribution trends of evaluation variables changed significantly. The contribution of physical capital stock was replaced by that of export rate in the western region, except Guangxi and Yunnan. When dynamic FDI performance declined in 2012, the contribution trend was relatively similar to that of the eastern region. This implies that the contribution of physical capital stock was replaced by that of export rate. From a provincial perspective, the contribution of export rate was particularly high in Inner Mongolia, Qinghai, Gansu, and Ningxia. However, the contribution scores of western provinces are differences in the distribution. For example, the contribution scores of Inner Mongolia, Chongqing, and Shaanxi were 0.8, 0.6, and 0.6, respectively. Whereas those of Ningxia and Guizhou were both 0.1. Thus, the control of evaluation variables in different provinces was relatively diverse. The western provinces have dual characteristics for FDI performance. Inner Mongolia, as a representative of the western region, has the advantages of a lower consumption level, more energy resources, and lower environmental thresholds, whereas non-negligible disadvantages are lower performance in FDI for a higher input demand.

## 4. Conclusions and Policy Implications

This paper evaluated the different dimensions of provincial dynamic FDI performance in China. One of the important contributions of this paper was to evaluate the inter-temporal effect of physical capital stock. First, an expanded dynamic Malmquist model with the inter-temporal effects was built based on dynamic SBM (Tone and Tsutsui 2010). Next, based on the thinking of traditional Malmquist decomposition (Färe et al. 1994), DTC is a decomposed index of dynamic Malmquist model. The empirical results revealed that DTC is the major bottleneck in dynamic FDI performance. This means the efficiency of inter-temporal influence is lower than other decomposed indices. This finding is consistent with those of previous studies (Yang et al. 2015; Belloumi and Alshehry 2018) on sustainability of economic development. Therefore, this paper is a valuable complementary study that empirically clarifies the direct influences of FDI performance. Furthermore, this paper establishes a critical bridge between FDI performance and sustainability expectation, although many previous studies (Hussain and Haque 2016; Dal Bianco and Loan 2017) have frequently linked FDI to its extrinsic

outcomes, such as profits or financial performance, without considering the inter-temporal effect. The main findings are as follows:

(1) DTC is a critical index in dynamic FDI performance in China. As the calculation results show, potential power improvement is a major step for FDI performance improvement. DTEC experienced a brief increase before 2000. Thereafter, the index had a decreasing trend until the last two years. The slight increase depends on the scale effect of evaluation variables.

(2) Scale effect is a bottleneck for dynamic FDI performance in the eastern region in China. FDI of the eastern region began after economic openness, which led to the continued predominance of resources. However, resource slack typically results in negligence of the scale effect. Nevertheless, the contribution scores of physical capital stock and export rate are considerable. Additionally, the contribution of physical capital stock is gradually replaced by that of export rate.

(3) Dynamic FDI performance is diverse in the western region. Because of the large gaps in provincial dynamic FDI performance in China, the western region faces the danger of "waste first, and then govern". Additionally, the contribution changes of evaluation variables are significant from both diachronic and regional perspectives. This means significant differences exist in the control management.

(4) Dynamic FDI performance in the midland gradually changes from mixed-indices-driven to DTC-driven. The improvement of dynamic FDI performance almost relies on auto-regulation. This indicates that the midland does not receive a free ride from the eastern region during FDI development. For the contribution scores of evaluation variables, the midland provinces have a stable trend. However, in contrast to the eastern region, the contribution scores of export rate in the midland is replaced by that of physical capital stock.

In conclusion, for a sustainability strategy, investors should surrender the profit in a short run to emphasize quality increase instead of quantity improvement. Then, investors can complete the identity change from foreign speculators to strategic investors. For decision makers, the pursuit of FDI growth with single minded and blind would result in the neglect of the potential opportunity for FDI. Additionally, FDI growth must not be pursued at the expense of environmental pollution. Therefore, the solution is incorporating sustainability into FDI to stimulate the spillover effect.

Considering the regional resource endowment, investors should capitalize on the opportunity to increase their potential market according to the differences in provincial dynamic FDI performance. At the macro-economic level, policy bonus must be comprehended. Based on the thinking of Nazarczuk and Krajewska (2018), the role of SEZs (special economic zones) in FDI attraction is very important, such as resource endowments, heterogeneous border effects, and the importance of close proximity to SEZs. For example, regional coordination as in the Beijing-Tianjin-Hebei Metropolitan Area, and free trade zone foundation as in the Shanghai Pilot Free Trade Zone and the Shenzhen Qianhai Free Trade Zone. At the micro-economic level, the transformation for an advantage province is a crucial objective, particularly for the provinces in the midland and western regions. For decision makers, the reorganization of FDI patterns is extremely urgent. Focusing on the development of the midland is a potentially effective strategy. Then midland can catch up with the eastern region. Additionally, investment policy should support the importation of knowledge to strengthen the potential power of FDI. For example, the "Talent introduction program" is popular in lots of provinces, such as Tianjin and Chengdu in the Sichuan province.

This paper has some implications based on the empirical results of FDI in China. However, several limitations should be addressed in the future. First, this study has considered the inter-temporal effect instead of a long run effect of physical capital stock. Further development and evidence from additional extensions of the Malmquist model and influence periods are required. Second, this paper has focused on the provincial FDI performance evaluation. For the reliability of data, the original data of provincial FDI is collected from the State Statistical Bureau of China. Since some provincial data in subsequent years is missing, the period of this paper is from 1998 to 2013. Therefore, reliable data

sources are a direction for further research. Third, because of the robustness of multicollinearity, the regression correlation model and difference analysis among cluster variables are necessary before the cluster analysis. For further tests of clustering results, a structural model can be introduced. Future studies will remedy these limitations to make more comprehensive contributions.

**Author Contributions:** Conceptualization, X.Z. (Xinna Zhao) and X.Z. (Xiaohong Zhang); Data curation, Y.T.; Formal analysis, M.L.; Funding acquisition, M.L.; Investigation, Y.T.; Methodology, X.Z. (Xinna Zhao) and Y.T.; Project administration, X.Z. (Xiaohong Zhang); Resources, M.L.; Software, X.Z. (Xinna Zhao); Supervision, X.Z. (Xiaohong Zhang); Validation, M.L. and X.Z. (Xinna Zhao); Visualization, Y.T.; Writing—original draft, X.Z. (Xinna Zhao) and Y.T.; Writing—review and editing, M.L. and X.Z. (Xiaohong Zhang).

**Funding:** This research was funded by National Natural Science Foundation of China, grant number 71703009, Social Science Fund Project of Beijing, grant number 16YJC049, 16YJC062, Scientific Project of Beijing Education Commission, grant number SM201710017001, National Students' project for innovation and entrepreneurship training program, grant number 2018J00225, 2019J00012. Great Wall Scholar of Beijing, grant number CIT&TCD20180314.

**Conflicts of Interest:** The authors declare no conflict of interest.

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
