# Peer review of "Foreign Direct Investment Dynamic Performance with Low-Carbon Influence: A Provincial Comparative Application in China"

_ijfs, doi:10.3390/ijfs7030046_

Round 1

Reviewer 1 Report

The review of the paper titled: „Foreign Direct Investment Dynamic Performance with Low-Carbon Influence: a provincial comparative application in China”:

General remarks:

-        The novelty of the paper is mediocre,

-        Introduction needs major review (see below for details),

-        The paper lacks theoretical review section and discussion of findings,

-        Its greatest weakness are lengthy descriptions of the data, which are then presented on figures with insufficient implications and potential reasons/causes of the  situation,

-        The literature section should be enlarged,  

-        The quality of the language style should be enhanced.

Abstract

- The text needs proofreading as it is flourished with style flaws: eg. line 3: Because -> because, line 18: productivity by M… model -> productivity with the use of, line 17: This paper evaluate -> The paper evaluates,…

- the abstract is organised in the wrong way as it does not provide information on the results and novelty of the paper

Introduction

-        It does not provide information on the novelty of the paper nor it sufficiently presents the motivation of undertaking the research. The authors should also present the state of the art.

-        At this stage the introduction is very chaotic and some parts of the introduction are not linked with each other ,

-        Frequent style glitches are present,

-        FDI importation – please do not use this term also in other parts of the paper,

-        At the end of the introduction the remainder of the paper should be presented,

-        At line 66: This paper presents… one should start with a new paragraph,

-        The presented literature review barely presents the grounds of undertaking the research.

2. Methodology

-        Authors should start with some introduction to the model, before they start its description. The text paragraphs could provide more information.

-        Some abbreviations/terms must be clearly explained in the text: DMU, DTEC, DTC.

-        Authors does not provide information on data sources, nor they comment on the problems with data or how they were calculated.

-        The table 1 and 2 are not introduced in the paper before the two tables appear in the text

-        Line 112-113: this reasoning is insufficient for undertaking the research

3. Empirical results

- This section needs major revision as it is full of lengthy descriptions, which are not necessary, since the authors reproduce  in the text what is visible on charts. They do however does not provide sufficient comments on the results. They should do it by showing the causes, implications, similarities with other research/models.

-It is worthwhile to add a map with location of the provinces in tables 3 (maybe in the appendix),

- The authors does not provide information on the specific method of clustering (section 3.2): (hierarchical clustering is a group of techniques). They do not provide information basing on what criteria they have chosen the four groups. It is advisable then to correct it. They do not provide information whether other clustering techniques were run to verify the stability of the results. The reader does not have either information on the variables that were used to the clustering.  For sure, the authors should thing of more descriptive names of these 4 groupings and should place them on figure 4. These four clusters should be presented on the map of China, since the international reader would probably have little knowledge of the geography of China. Additional table presenting mean values of particular clusters would enhance the clarity of this section.

- the section does not provide the discussion of findings, references of the results to other papers

- one could better introduce the reasoning of changing the scope of the research to eastern, central , western regions.

Conclusion:

-        Lack of the discussion of findings/implications with other research papers (they should be cited)

-        The authors could extend the policy implications part by showing the lesson for regional authorities.

Author Response

Overall Response: We thank our reviewer for the diligence in reviewing our manuscript to provide constructive comments for improvement. We also appreciate the encouragement. We have incorporated the suggestions in the revised manuscript.  We believe that the constructive comments have helped us improve this manuscript significantly to reflect the goals and objectives of International Journal of Finance Studies.

Point 1: General remarks: The novelty of the paper is mediocre. Introduction needs major review (see below for details). The paper lacks theoretical review section and discussion of findings. Its greatest weakness are lengthy descriptions of the data, which are then presented on figures with insufficient implications and potential reasons/causes of the situation. The literature section should be enlarged. The quality of the language style should be enhanced.

Response 1: For the further revision, we revised the manuscript such as English language and style, explanations of the introduction and details of result analysis. The definite explanations are shown as Point 2 to Point 6.

Again, we sincerely appreciate your encouragement and suggestions for improvement.

Point 2: Abstract

(1)  The text needs proofreading as it is flourished with style flaws: eg. line 3: Because -> because, line 18: productivity by M… model -> productivity with the use of, line 17: This paper evaluate -> The paper evaluates,…

(2)  the abstract is organised in the wrong way as it does not provide information on the results and novelty of the paper

Response 2: For the further description of Abstract, we revised the manuscript such as English language and style, explanations of the empirical results and novelty of this paper.

(1) First, we have corrected “Because” to “because” on line 16. Next, “This paper evaluate” has been corrected to “The paper evaluates” on line 19. Then, “productivity by Malmquist model” has been corrected to “productivity with the use of dynamic Malmquist model” on line 19-20. Last, we have revised the abstract section. For example, we have added “of” on line 12. We also change “of” to “in” on line 13.

(2) For interpreting the empirical results, we have revised as “The empirical results reveal that dynamic variable is the bottleneck for most Chinese provinces in FDI performance. It is only efficient in few municipalities and provinces, such as Shanghai and Guangdong. And the energy conservation is more efficient for performance evaluation in Chinese east region.” on line 23-26.

For interpreting the novelty of this paper, we have emphasized the inter-temporal influence as “Differ with traditional evaluation, this paper focuses on the inter-temporal influence in FDI performance evaluation. To understand the inter-temporal effects, physical capital stock is defined as dynamic variables in FDI sustainability performance. Besides that, with the pressure to reduce emissions, energy consumption is also considered during the evaluation.” on line 20-23.

Point 3: Introduction

(1)  It does not provide information on the novelty of the paper nor does it sufficiently present the motivation of undertaking the research. The authors should also present the state of the art.

(2)  At this stage the introduction is very chaotic and some parts of the introduction are not linked with each other,

(3)  Frequent style glitches are present,

(4)  FDI importation – please do not use this term also in other parts of the paper,

(5)  At the end of the introduction the remainder of the paper should be presented,

(6)  At line 66: This paper presents… one should start with a new paragraph,

(7)  The presented literature review barely presents the grounds of undertaking the research.

Response 3: For the further description of Introduction, we revised the manuscript such as English language and style, explanations of the literatures and novelty of this paper.

(1) For the further description of the purpose in the introduction section, we have revised the purpose of this paper as “Therefore, the purpose of this paper focuses on not only the traditional performance of provincial FDI in China, but also considers energy influence with the thinking of low-carbon economy. Moreover, physical capital stock as the inter-temporal effect will make the dynamic performance more generic.” on line 47-49.

For the contribution of this study, we have reported it in the introduction section as “This paper makes contributions at the methodological and the application levels. First, by incorporating the inter-temporal effect into Malmquist model, this paper facilitates explaining the sustainability of participating in provincial FDI considering the physical capital stock as the carry-over activities. Carry-over activities are defined as the special variables which not only take effect in one period, but also have a long-term effect in multiple periods. Second, considering the low-carbon economy, this paper evaluates FDI performance with the energy influence. Third, this paper investigates the different contributions of influence factors to balance the FDI regional differences. Besides that, this paper also has practical implications; the findings provide decision makers with an increasing understanding of the FDI sustainable development mechanism and investors with a reference for distributing their FDI.” on line 84-93.

(2) We have revised the introduction section. For example, we have added as “Thus, FDI is essential for economic development in China.” to connect the context on line 35-36. Moreover, we have added as “Thus, the FDI in China is disproportionately distributed. Besides that, similar with the development track of Chinese economy, FDI quality is more essential than the pure quantity.” to connect the context on line 42-43.

(3) We have revised the introduction section. For example, we have revised as “Comparing with last year, it increases 69.8%. FDI from UK increases 150.1%. FDI from Germany increases 79.3%. And that of United States also increases 7.7%.” on line 38-39. Moreover, we have corrected “Because” to “because” on line 40.

(4) We have changed all “FDI importation” in this paper.

(5) For the further description of the steps in the introduction section, we have revised the paper as “This paper is structured as follows. First, a conceptual dynamic Malmquist model is constructed and decomposed. The measurements of the FDI performance among provinces are presented in Section 2. Then, an example of provincial FDI in China with a more dynamic viewpoint is discussed using cluster analysis and variable contribution analysis in Section 3. Last, the conclusions as well as the suggestions and limitations are presented in the last section.” on line 94-98.

(6) We have revised “This paper presents… ” on line 84 as a start with a new paragraph.

(7) We have rewritten the literature review section.

First, we have changed the tense from present to past in the literature review. For example, “A literature review revealed the importance of FDI performance (Rashid and Lin, 2018).” on line 50 and “Kueh (1992) discussed the reasons for FDI distribution diversity and the implications of policy supports. The results suggested that FDI distribution was changing from the southeast to north and northeast coasts as well as in the midland and west region.” on line 56-59.

Second, we have added the literatures about the influence variables of FDI as “In the context of a stabilized economy and increasing FDI, previous literatures focused on the influence variables of FDI, such as the degree of openness (Walter and Ugelow, 1969), domestic investment (Cheng and Kwan, 2000), the labor force (Lei et al., 2013), and the allocation of resources (Rashid and Lin, 2018).” as line 60-63.

Third, we have added the literatures about the background of our model as “The input and output variables in these literatures are always selected to reflect the cost sources as the inputs and the revenue sources as the outputs (Feng and Serletis, 2010; Portela and Thanassoulis, 2010; Jiang and He, 2018). However, some special variables not only take effect in one period, but also have a long-term effect, such as deposit in banks and fixed assets in companies (Bagchi et al., 2019). As Zhao and Zhong (2017) pointed out, traditional Malmquist productivity index does not account for the effect of these special variables between consecutive periods. This can lead the performances to over-incentive in the short-term and under-incentive in the long-term. Dynamic DEA model was proposed for dealing with inter-connecting activities by Färe and Groffkopf (1996). Tone and Tsutsui (2010) extended the dynamic DEA model by adding carry-over effects among different periods in the framework of the SBM model, called Dynamic SBM (DSBM). For the inter-temporal effect of multiple period, this paper introduces the dynamic thinking from DSBM into traditional MPI and builds a dynamic Malmquist productivity index.” on line 72-83.

Point 4: Methodology

(1)  Authors should start with some introduction to the model, before they start its description. The text paragraphs could provide more information.

(2)  Some abbreviations/terms must be clearly explained in the text: DMU, DTEC, DTC.

(3)  Authors does not provide information on data sources, nor they comment on the problems with data or how they were calculated.

(4)  The table 1 and 2 are not introduced in the paper before the two tables appear in the text

(5)  Line 112-113: this reasoning is insufficient for undertaking the research

Response 4: For the further description of Methodology, we revised the manuscript such as English language and style, explanations of the abbreviations and details of model.

(1) For interpreting the model, we have added some introduction of the model, such as “Before describing the detailed formulations, the conceptual framework of dynamic Malmquist model is shown in Figure 1. Provinces in China are defined as the decision-making units (DMUs) which are the research objects during FDI performance evaluation.” on line 101-103.

(2) We have added the explanations of the abbreviations. For example, DMU is decision-making unit as “Provinces in China are defined as the decision-making units (DMUs) which are the research objects during FDI performance evaluation.” on line 102-103. Moreover, we have added the explanations of DTEC and DTC as “The method of dynamic Malmquist model decomposition follows that of traditional Malmquist model. The traditional one is extended into the new model considering the carry-over activities. Thus, dynamic Malmquist can be composed into overall efficiency change (DTEC) and dynamic technical change (DTC). Similar with traditional Malmquist decomposition thinking (Färe et al, 1994), DTEC means the hard power of performance improvement, such as the allocation of evaluation variables and scale efficiency. And DTC is denoted as the soft power, such as the managerial style.” on line 124-129.

(3) For the data of the paper, we revised the manuscript. For example, we have added the further explanation of variables, in the “2.2. Data Explanation” section. Such as “Previous studies of performance evaluation always focused on physical capital stock, human capital stock and output value based on the economic growth theory.” on line 131-132.

For further description, we have added the source of the data as “Similar with most of previous literatures, average FDI is defined as output variables. Average human capital stock, average energy consumption, and export rate are defined as input variables. Considering the hysteresis of physical capital stock, dynamic variable is defined based on average physical capital stock. For simple calculation, the inter-temporal effect of physical capital stock focuses on the adjacent period.” on line 138-142. Moreover, “All the data is from the official statistics such as China statistical yearbook, China city statistical yearbook, and China energy statistical yearbook.” on line 166-167.

And then, for the presentation of variables’ definition, we have added as “The definition of variable are shown as follow.

① Physical capital stock: On the thinking of the perpetual inventory method (Zhang et al., 2004). First, the current fixed capital formation is defined as the current investment. Then, the basic period is defined as the 10 times of fixed capital formation at 1952. At last, depreciation rate as 10.96% is used during the calculation based on Lei (2009).

② Human capital stock: Educated manpower is the main inducement of FDI competition. On the thinking of Zhao (2009), this paper denotes illiteracy as 3, primary education as 6, junior high school education as 9, high school education as 12, college education and above as 16. The sum of the indices is defined as the human capital stock.

③ Energy consumption: The energy cost and availability is the main factor considering the low-carbon economic development. For some data missing, the average of adjacent provinces data is offered instead of the Hainan data at 2002, the Hunan province data at 1997 and 1998.

④ Export rate: The economic openness is the key factor for the multinationals investment. The regional openness is the basic of FDI performance. So the export rate is defined as the openness benchmark.

⑤ Foreign directed investment: FDI is defined as a production function. If there are the same inputs, the more funds express the more investment benefits.” on line 142-158.

(4) For interpreting the Table 1, we have added some introduction, such as “For eliminating the influence of population differences among provinces, the average data is used in this paper. For avoiding the multicollinearity problem, the correlation coefficients of all evaluation variables are listed in Table 1 as below.” on line 159-161.

For interpreting the Table 2, we have added some introduction, such as “All the data is from the official statistics such as China statistical yearbook, China city statistical yearbook, and China energy statistical yearbook. As shown in Table 2, the descriptive statistics results of the data are expressed as the average, standard deviation, minimum, and maximum. Significant differences.” on line 166-169.

(5) For explaining the reasoning for undertaking the research, we have revised as “As shown in Table 2, input variables exhibit minimal differences. However, the differences among FDI data are famous. These results indicate that FDI dynamic performance exhibits significant differences among provinces during relatively fixed inputs. Therefore, it is necessary to examine the bottlenecks in FDI dynamic performance improvement. So, the rest of the paper is a thorough analysis on the differences of FDI performances among provinces.” on line 171-175.      

Point 5: Empirical results

(1)  This section needs major revision as it is full of lengthy descriptions, which are not necessary, since the authors reproduce  in the text what is visible on charts. They do however does not provide sufficient comments on the results. They should do it by showing the causes, implications, similarities with other research/models.

(2)  It is worthwhile to add a map with location of the provinces in tables 3 (maybe in the appendix),

(3)  The authors does not provide information on the specific method of clustering (section 3.2): (hierarchical clustering is a group of techniques). They do not provide information basing on what criteria they have chosen the four groups. It is advisable then to correct it. They do not provide information whether other clustering techniques were run to verify the stability of the results. The reader does not have either information on the variables that were used to the clustering.  For sure, the authors should thing of more descriptive names of these 4 groupings and should place them on figure 4. These four clusters should be presented on the map of China, since the international reader would probably have little knowledge of the geography of China. Additional table presenting mean values of particular clusters would enhance the clarity of this section.

(4)  the section does not provide the discussion of findings, references of the results to other papers

(5)  one could better introduce the reasoning of changing the scope of the research to eastern, central , western regions.

Response 5: For the further description of Empirical results, we revised the manuscript such as English language and style, explanations of the clusters and details of results.

(1) For further interpreting the causes, implications, similarities with other research/models, we have rewritten the empirical results. For example, we have added some explanations after the analysis of four clusters.

After cluster one analysis, we have added as “With a leading position in terms of allocation efficiency, the main evaluative criterion presents profit growth. So the provincial FDI of this cluster are similar with the Dog group. In that way, the strategy of this cluster is stabilization development. The most efficient breakthrough for improving performance is keeping the advantage of profit growth in the traditional evaluation. In order to earn more profit, this strategy is to give full advantage of inter-temporal activity cycles. This influence cannot be ignored.” on line 288-293.

After cluster two analysis, we have added as “With the obvious advantage of soft power, the strategy of this cluster is expansion. The most efficient breakthrough strategy for improving performance is actively improving scale efficiency growth and market opportunity with the goal of improving long-term performance. In that way, this cluster can improve its performance in order to strengthen its leading position as a typical Star group. Considering the sustainable development, on the one hand, this group should evaluate the inter-temporal efficiency changes of carry-over activity (physical capital stock) in the long-term development. On the other hand, the cluster should continue to consider hard power strengthen in order to remedy the finance data without considering the completeness of outputs. Above all, these traditional advantages combined with a sustainable development view could keep this cluster in the top position.” on line 301-310.

After cluster three analysis, we have added as “The strategy of this cluster should be retrenchment. This means they should gather strength to break through in order to avoid becoming a typical Dog group. The most efficient breakthrough for improving performance in this cluster is the control of carry-over activities. This means that strengthened management of carry-over activity (physical capital stock) can enhance the dynamic FDI performance. The specific method is incorporating the inter-temporal influence into current incentive mechanisms. The embodiment is that evaluation indicators transform from financial data at one period into inter-temporal relative data, such as inter-temporal benefit increasing rate, average daily balance and contribution. This breakthrough can not only increase the advantage, but also match the current sustainable development strategy.” on line 320-328.

After cluster two analysis, we have added as “With the advantage position for the scale efficiency of FDI variables. This sets a better example for other cluster. However, they still have problems that need to be solved urgently with the traditional profit competition. The strategy of this group is reinforcement. The most efficient breakthrough strategy for improving the performance of this group is modernizing the management to develop profit markets rapidly to catch up with competitors. For example, using think-tank-management-mode transforms traditional profit benefit into traditional profit efficiency for a higher traditional index, especially TC and PTC. If this group systematically selects strategies to pursue profit growth according to its strength and advantage, it will turn to the Star group.” on line 338-345.

(2) For explaining the distribution of provincial performance, we have added a map with location of Chinese provinces as shown in Figure 4 on line 230.

(3) For emphasizing the regional heterogeneity, we have added the explanation of the clustering analysis as “To identify the peer effects of interprovincial in FDI dynamic performance, the provinces are classified using hierarchical clustering. For the previous calculation by dynamic Malmquist model, the different performance indices of provincial FDI stand for their own regional characteristics. Therefore, three decomposed efficiency indices (DTC, DSEC, and DPTC) with the regional heterogeneity are the basics for the cluster analysis. And then, this paper tests several other methods including the Ward-method based on the theory of hierarchical cluster analysis. Only furthest neighbor distance method can classify the discrimination of provincial FDI performance. For example, based on the Ward-method, cluster one includes more than one half of Chinese provinces which the number is sixteen. Both of cluster two and cluster four only include three provinces. So the furthest neighbor distance method is more suitable for this empirical research comparing with other methods. Based on the provincial regional characteristics, this paper classifies the provinces into four clusters.” on line 250-260.

Then, we have changed Figure 5 to a map which presents four clusters as shown on line 257. Moreover, we have revised the description as “The provinces are classified into four clusters as shown in Figure 4. The west provinces comprise cluster Two and cluster Three. The midland provinces comprise cluster Three and cluster Four. And the east provinces comprise cluster One and cluster Two. The regional influence is significant for the decomposing efficiency indices of FDI performance.” on line 261-264.

Last, we have added Table 4 to presenting mean values of the four clusters as shown on line 262. And we have also added some explanations as “Basing on cluster analysis, the average decomposed indices and the corresponding rankings are presented in Table 4. The soft power effect is denoted by DTC. The allocation effect is denoted by DPTC. And the scale effect is denoted by DSEC.” on line 267-269.

(4) For the discussion of findings, we have revised in the conclusion. For example, “DTC is critical in FDI dynamic performance in China. Therefore, soft power improvement is a major step for FDI. DTEC experiences brief increasing before 2000. Thereafter, the index declines until the past 2 years.” on line 434-436. “For the contributions of variables, the trends of midland provinces are flatten. However, in contrast to east region, the contribution of the export rate in midland is replaced by that of capital.” on line 449-450.

We have added some literatures to compare the findings with others in the conclusion. For example, “This finding is consistent with those of previous studies (Yang, Wan, and Ma, 2015; Belloumi and Alshehry, 2018) on economic sustainability.” on line 427-428. And “Although many previous studies (Hussain and Haque, 2016; Silvia et al., 2017) have frequently linked FDI to its extrinsic outcomes, such as profits or financial performance, without considering the inter-temporal effect.” on line 430-433.

(5)  For the further interpreting the research to east, midland, and west in China, we have added some explanations in empirical analysis section. For example, “As shown in Figure 4, the deeper color province indicates the higher FDI performance. The different color provincial boundary shows the different regions. The white boundary is midland in China. The brighter boundary is west province in China. And the deeper one is the east in China.” on line 222-225.

Point 6: Conclusion

(1)   Lack of the discussion of findings/implications with other research papers (they should be cited)

(2)  The authors could extend the policy implications part by showing the lesson for regional authorities.

Response 6: For the further description of Conclusion, we revised the manuscript such as English language and style, explanations of the findings and details of implications.

(1) We have added some literatures to compare the findings with others in Conclusion. For example, “First, an expanded dynamic Malmquist model with the inter-temporal effects is proposed based on the dynamic SBM (Tone and Tsutsui, 2010). Next, based on the thinking of traditional Malmquist decomposition (Färe et al., 1994), DTC is a decomposition of dynamic Malmquist.” on line 422-425. “This finding is consistent with those of previous studies (Yang, Wan, and Ma, 2015; Belloumi and Alshehry, 2018) on economic sustainability.” on line 427-428. And “Although many previous studies (Hussain and Haque, 2016; Silvia et al., 2017) have frequently linked FDI to its extrinsic outcomes, such as profits or financial performance, without considering the inter-temporal effect.” on line 430-433.

(2) For interpreting the policy implications, we have added some explanations, such as “Considering the local resource endowment, investors should capitalize on the opportunity to increase their potential market according to the differences in FDI dynamic performance among provinces. At the macro level, policy support guidance must be comprehended. For example, regional coordination as beijing-tianjin-hebei metropolitan area, free trade zone foundation as Shanghai Pilot Free Trade Zone and Shenzhen Qianhai Free Trade Zone.” on line 457-461.

Thank you again.  We appreciate your encouragement and recommendation.

Reviewer 2 Report

I think the paper is interesting, however, there are a few things I think needs improvement. 

I think you need to pay attention to the quality of language and expression. Although, I must acknowledge that it is good enough to convey the message but still you can go through the paper to rephrase certain sentences.  There is seems no coherence, for instance, under data explanation you wrote"Hysteresis of physical capital stock exists in economic growth and technical progress (He, 2001). Furthermore, Hou (2016) proved the objectivity of physical capital stock hysteresis. He also demonstrated that GDP growth depends on the previous year’s fixed capital investment". What are you talking about, the data section should explain your data with no other studies. Similarly, other parts of the paper are not coherent either.  The presentation and structure could also be improved, for instance, you did not say much about what data you used. There should be clear discussion and explanation of variables.  I gathered that you might have used data from 1998 to 2013, this is 2019. I would suggest you update your data set to the latest possible dates.  

Author Response

Overall Response: We thank our reviewer for the diligence in reviewing our manuscript to provide constructive comments for improvement. We also appreciate the encouragement. We have incorporated the suggestions in the revised manuscript.  We believe that the constructive comments have helped us improve this manuscript significantly to reflect the goals and objectives of International Journal of Finance Studies.

Point 1: I think you need to pay attention to the quality of language and expression. Although, I must acknowledge that it is good enough to convey the message but still you can go through the paper to rephrase certain sentences. 

 Response 1: For the further revision, we revised the manuscript such as English language and style, explanations of the evaluation variables and details of result analysis.

Again, we sincerely appreciate your encouragement and suggestions for improvement.

Point 2: There is seems no coherence, for instance, under data explanation you wrote "Hysteresis of physical capital stock exists in economic growth and technical progress (He, 2001). Furthermore, Hou (2016) proved the objectivity of physical capital stock hysteresis. He also demonstrated that GDP growth depends on the previous year’s fixed capital investment". What are you talking about, the data section should explain your data with no other studies.

Response 2: For the coherent description, we have revised the explanation of “physical capital stock” as “Previous studies of performance evaluation always focused on physical capital stock, human capital stock and output value based on the economic growth theory.” on line 131-132.

For further description, we have added the explanation of the variable as “Considering the hysteresis of physical capital stock, dynamic variable is defined based on average physical capital stock. For simple calculation, the inter-temporal effect of physical capital stock focuses on the adjacent period.” on line 140-142.

And then, we have offered the definition of physical capital stock as “On the thinking of the perpetual inventory method. First, the current fixed capital formation is defined as the current investment. Then, the basic period is defined as the 10 times of fixed capital formation at 1952. At last, depreciation rate as 10.96% is used during the calculation based on Lei (2009).” on line 143-146.

Point 3: Similarly, other parts of the paper are not coherent either. The presentation and structure could also be improved, for instance, you did not say much about what data you used. There should be clear discussion and explanation of variables. 

Response 3: For the coherence of the paper, we revised the manuscript. For example, we have added the further explanation of variables, in the “2.2. Data Explanation” section. Such as “Previous studies of performance evaluation always focused on physical capital stock, human capital stock and output value based on the economic growth theory.” on line 131-132.

For further description, we have added the source of the data as “Similar with most of previous literatures, average FDI is defined as output variables. Average human capital stock, average energy consumption, and export rate are defined as input variables. Considering the hysteresis of physical capital stock, dynamic variable is defined based on average physical capital stock. For simple calculation, the inter-temporal effect of physical capital stock focuses on the adjacent period.” on line 138-142.  “For eliminating the influence of population differences among provinces, the average data is used in this paper.” On line 159-160. And “All the data is from the official statistics such as China statistical yearbook, China city statistical yearbook, and China energy statistical yearbook.” on line 166-167.

And then, for the presentation of variables’ definition, we have added as “The definition of variable are shown as follow.

(1) Physical capital stock: On the thinking of the perpetual inventory method. First, the current fixed capital formation is defined as the current investment. Then, the basic period is defined as the 10 times of fixed capital formation at 1952. At last, depreciation rate as 10.96% is used during the calculation based on Lei (2009).

(2) Human capital stock: Educated manpower is the main inducement of FDI competition. On the thinking of Zhao (2009), this paper denotes illiteracy as 3, primary education as 6, junior high school education as 9, high school education as 12, college education and above as 16. The sum of the indices is defined as the human capital stock.

(3) Energy consumption: The energy cost and availability is the main factor considering the low-carbon economic development. For some data missing, the average of adjacent provinces data is offered instead of the Hainan data at 2002, the Hunan province data at 1997 and 1998.

(4) Export rate: The economic openness is the key factor for the multinationals investment. The regional openness is the basic of FDI performance. So the export rate is defined as the openness benchmark.

(5) Foreign directed investment: FDI is defined as a production function. If there are the same inputs, the more funds express the more investment benefits.” on line 142-158.

Point 4: I gathered that you might have used data from 1998 to 2013, this is 2019. I would suggest you update your data set to the latest possible dates.  

Response 4: For the limitations of the data continuity in conclusion section, we have revised the limitations of this paper as “Second, this paper has focused on the provincial FDI performance evaluation. For the reliability of data, the original data of provincial FDI is collected from the State Statistical Bureau of China. However, the research period is from 1998 to 2013 for the missing of some provincial data in subsequent years. Therefore, reliable data sources is a direction for the further research.” on line 470-474.

Thank you again.  We appreciate your encouragement and recommendation.

Round 2

Reviewer 1 Report

Review of the paper titled ”Foreign Direct Investment Dynamic Performance 2 with Low-Carbon Influence: A Provincial 3 Comparative Application in China” – rev 1:

- the authors have greatly improved the paper in accordance to my suggestions – I would like to thank for that;

- my general opinion on the paper is positive, however prior to publication there are a few minor issues left for correction:  

- page 4, line 132: Previous studies of performance evaluation -> Previous studies on …

- page 4, line 135: as output variables. -> as output variable

- page 4., line 143: The definition of variable are shown as follow. -> The definition of variables …

- page 5, lines 155-157, the authors  could refer to https://doi.org/10.2478/bog-2018-0007

- the name of table 1 could be more precise: Correlation coefficients of factors. -> Correlation coefficients of factors.

- please indicate significant correlations by asterisk in table 1, note that correlation of 0.750 is a high one. Normally, one would expect correlations lower than 0.5

- name of the table 2 is ambiguous. One could try with “Descriptive statistics of the variables used in the study”

- normally you don’t put a dot after the table’s name

- shouldn’t “Figure 4. Dynamic performance distribution in China” should be more clear? The next sentence could be rewritten to be more comprehend.

- page 10, line 403: for other cluster. -> for other clusters.

- page 9, line 292: what the authors mean by the “Dog group”?

- the authors should find better collocations other than soft power, hard power, which seem to be vague

- page 13, line 425: study is to evaluating the impact -> study is to evaluate the impact

- page 13, line 438: govern.” -> govern”.

- page 14, line 464: beijing-tianjin-hebei metropolitan area -> shouldn’t it be written from capital letters?

- page 14, line 469: easte region. -> eastern region

- page 14, line 470: term „soft power.” Is misleading, please be more clear.

- page 14, line 477: research period is from 1998 to 2013 for the missing of some provincial data -> the sentence is not clear

- page 14, line 481: model can be introduce. -> can be introduced.

- in the concluding part authors could refer more to the role of SEZs in FDI attraction, eg. https://doi.org/10.24136/eq.2018.004 , similarly in the introductory part local factors attracting FDI could be slightly underlined.

Author Response

Response to Reviewer 1 Comments

Overall Response: We thank you for the diligence in reviewing our manuscript to provide constructive comments for improvement. We also appreciate the encouragement. We have incorporated the suggestions in the revised manuscript. We believe that the constructive comments have helped us improve this manuscript significantly to reflect the goals and objectives of International Journal of Finance Studies.

Point 1: Review of the paper titled “Foreign Direct Investment Dynamic Performance with Low-Carbon Influence: A Provincial Comparative Application in China” – rev 1: the authors have greatly improved the paper in accordance to my suggestions – I would like to thank for that; my general opinion on the paper is positive, however prior to publication there are a few minor issues left for correction: 

 Response 1: For the further description, we have revised the manuscript such as English language and style, explanations of the evaluation variables and details of empirical analysis.

Again, we sincerely appreciate your encouragement and suggestions for improvement.

Point 2: page 4

line 132: Previous studies of performance evaluation -> Previous studies on …

line 135: as output variables. -> as output variable.

line 143: The definition of variable are shown as follow. -> The definition of variables …

Response 2: We have replaced “Previous studies of performance evaluation” with “Previous studies on performance evaluation” on line 136. Then, “as output variables” have been modified as “as output variable” on line 143. Last, “The definition of variable are shown as follow.” have been changed into “The definitions of variables are shown as follow.” on line 147.

Point 3: page 5, lines 155-157, the authors  could refer to https://doi.org/10.2478/bog-2018-0007

Response 3: We have added the literature for the reference of variables explanation as “based on Nazarczuk and Umiński (2018).” on line 161.

Point 4: Tables

- the name of table 1 could be more precise: Correlation coefficients of factors. -> Correlation coefficients of factors.

- please indicate significant correlations by asterisk in table 1, note that correlation of 0.750 is a high one. Normally, one would expect correlations lower than 0.5

- name of the table 2 is ambiguous. One could try with “Descriptive statistics of the variables used in the study”

- normally you don’t put a dot after the table’s name

Response 4: We have revised the information of tables as the followings. First, we have changed the name of table 1 as “Correlation coefficients of evaluation variables” on line 167.

Next, we have added the asterisk for the significant correlations in table 1 on line 168.

Then, the name of table 2 have been changed as “Descriptive statistics of the variables used in the study” on line 176.

Last, we have deleted the dots after the names of table 1, table 2, table 3, and table 4 on line 167, line 176, line 251, and line 278.

Point 5: shouldn’t “Figure 4. Dynamic performance distribution in China” should be more clear? The next sentence could be rewritten to be more comprehend.

Response 5: For more clear, we have revised figure 4 and figure 5 on line 236 and line 272. Furthermore, we have rewritten the next sentence of figure 4 as “The first three advantaged provinces and the last three disadvantaged provinces are shown in Table 3 according to their efficiency indices.” on line 238-239.

Point 6: page 10, line 403: for other cluster. -> for other clusters.

Response 6: we have changed “for other cluster” to “for other clusters” on line 347.

Point 7: page 9, line 292: what the authors mean by the “Dog group”?

Response 7: we have added the explanation for the cluster analysis as “In the spirit of the BCG Matrix, the clusters can be classified as Dog group, Question Mark group, Cash Cow group and Star group.” on line 276-277.

Point 8: the authors should find better collocations other than soft power, hard power, which seem to be vague

Response 8: For further explanation, we have changed “soft power” into “potential power” in the paper. And then, we have changed “hard power” into “attractive power” in the paper.

Point 9: page 13,

line 425: study is to evaluating the impact -> study is to evaluate the impact

line 438: govern.” -> govern”

Response 9: we have revision the mistakes as “study is to evaluate the impact” on line 429 and “waste first, and then govern” on line 453.

Point 10: page 14,

line 464: beijing-tianjin-hebei metropolitan area -> shouldn’t it be written from capital letters?

line 469: easte region. -> eastern region

line 470: term „soft power.” Is misleading, please be more clear.

line 477: research period is from 1998 to 2013 for the missing of some provincial data -> the sentence is not clear

line 481: model can be introduce. -> can be introduced.

 Response 10: we have changed the mistakes as following.

“beijing-tianjin-hebei metropolitan area” has been changed into “Beijing-Tianjin-Hebei metropolitan area” on line 471.

“easte region” has been changed into “eastern region” on line 476.

For further explanation, we have changed “soft power” into “potential power”. Then, we have added an example as “For example, “Talent introduction programme” is popular in lots of provinces such as Tianjin and Chengdu in Sichuan province.” on line 477-478.

For further explanation, we have modified as “Since, the missing of some provincial data in subsequent years, the period of this paper is from 1998 to 2013.” on line 485.

“model can be introduce.” has been changed into “model can be introduced.” on line 489.

Point 11: in the concluding part authors could refer more to the role of SEZs in FDI attraction, eg. https://doi.org/10.24136/eq.2018.004 , similarly in the introductory part local factors attracting FDI could be slightly underlined.

Response 11: For interpreting the concluding part, we have added some explanation of the SEZs in FDI attraction as “On the thinking of Nazarczuk and Krajewska (2018), the role of SEZs in FDI attraction is very important, such as resource endowments, heterogeneous border effects, and the importance of close proximity to special economic zones.” on line 468-471. And then, we have added some explanation in the introductory part as “Thus, FDI in China is disproportionately distributed. Similar with some other counties, such as Poland, the regional differences originate from the resource endowment, path dependency on the location of the industry and big city agglomerations, infrastructure quality, different economic prospects of potential cooperation with neighboring countries or transformation challenges (Nazarczuk and Umiński, 2018). Besides that, similar with the development track of Chinese economy, FDI quality is more essential than the pure quantity, especially in special economic zones (SEZs).” on line 42-47.

Response: Sorry for the mistakes. In detail, we have corrected the mistake as above, but not limited to the above.

For example, “is defined as dynamic variables” has been changed into “is defined as dynamic variable” on line 22.

We have changed “the provincial FDI of this cluster are similar” into “the provincial FDIs of this cluster are similar” on line 297.

We have modified “With the advantage position for the scale efficiency of FDI variables. This sets a better example for other cluster.” as “With the advantage position for the scale efficiency of FDI variables, this sets a better example for other clusters.” on line 346-347.

We have replaced “Future studies will develop these limitations” with “Future studies will remedy these limitations” on line 489.

Thank you again.  We appreciate your encouragement and recommendation.

Reviewer 2 Report

Your wring is still very weak. For example, "Future studies will develop these limitations" why should the devleop limitations ? this is just one example of many errors. 

Author Response

Response to Reviewer 2 Comments

Overall Response: We thank you for the diligence in reviewing our manuscript to provide constructive comments for improvement. We also appreciate the encouragement. We have incorporated the suggestions in the revised manuscript. We believe that the constructive comments have helped us improve this manuscript significantly to reflect the goals and objectives of International Journal of Finance Studies.

Point 1: Your wring is still very weak. For example, "Future studies will develop these limitations" why should the devleop limitations ? this is just one example of many errors.

Response 1: For the further description, we have revised the manuscript such as English language and style, explanations of the evaluation variables and details of empirical analysis. For example, we have replaced “Future studies will develop these limitations” with “Future studies will remedy these limitations” on line 489.

Again, we sincerely appreciate your encouragement and suggestions for improvement.

Sorry for the mistakes. In detail, we have corrected the mistake as above, but not limited to the above.

(1) For example, “is defined as dynamic variables” has been changed into “is defined as dynamic variable” on line 22.

(2)We have replaced “Previous studies of performance evaluation” with “Previous studies on performance evaluation” on line 136.

(3)  “as output variables” have been modified as “as output variable” on line 143.

(4)  “The definition of variable are shown as follow.” have been changed into “The definitions of variables are shown as follow.” on line 147.

(5) For more clear, we have revised figure 4 and figure 5 on line 236 and line 272.

(6) We have rewritten the sentence as “The first three advantaged provinces and the last three disadvantaged provinces are shown in Table 3 according to their efficiency indices.” on line 238-239.

(7) We have changed “the provincial FDI of this cluster are similar” into “the provincial FDIs of this cluster are similar” on line 297.

(8) We have modified “With the advantage position for the scale efficiency of FDI variables. This sets a better example for other cluster.” as “With the advantage position for the scale efficiency of FDI variables, this sets a better example for other clusters.” on line 346-347.

(8) We have revision the mistakes as “study is to evaluate the impact” on line 429 and “waste first, and then govern” on line 453.

(9) “beijing-tianjin-hebei metropolitan area” has been changed into “Beijing-Tianjin-Hebei metropolitan area” on line 471.

(10) “easte region” has been changed into “eastern region” on line 476.

Thank you again.  We appreciate your encouragement and recommendation.
